# KNOWLEDGE DISTILLATION WITH PERTURBED LOSS: FROM A VANILLA TEACHER TO A PROXY TEACHER

## ABSTRACT

Knowledge distillation is a popular technique to transfer knowledge from large teacher models to a small student model. Typically, the student learns to imitate the teacher by minimizing the KL divergence of its output distribution with the teacher's output distribution. In this work, we argue that such a learning objective is suboptimal because there exists a discrepancy between the teacher's output distribution and the ground truth label distribution. Therefore, forcing the student to blindly imitate the unreliable teacher output distribution leads to inferior performance. To this end, we propose a novel knowledge distillation objective *PTLoss* by first representing the vanilla KL-based distillation loss function via a Maclaurin series and then *perturbing* the leading-order terms in this series. This perturbed loss implicitly transforms the original teacher into a *proxy teacher* with a distribution closer to the ground truth distribution. We establish the theoretical connection between this "distribution closeness" and the student model generalizability, which enables us to select the PTLoss's perturbation coefficients in a principled way. Extensive experiments on multiple natural language processing tasks demonstrate the effectiveness of PTLoss with teachers of different scales.

## 1 INTRODUCTION

Knowledge distillation (KD) is a widely-used technique to transfer knowledge from large teacher models into a much smaller student model with minimum sacrifice of teacher model's predictive power (Buciluǎ et al., 2006; Hinton et al., 2015). The vanilla training objective in KD such as KL loss (Hinton et al., 2015; Menon et al., 2021; Stanton et al., 2021) encourages the student's outputs to be close to the teacher's outputs as much as possible, which implicitly assumes the teacher's outputs on the distillation data are perfect. However, the teacher's output distributions can be biased from the ground truth due to various factors, such as the inductive bias encoded in the teacher model architecture, miscalibration in the training procedure (Menon et al., 2021), or the bias in the teacher model training set (Liu et al., 2021; Lukasik et al., 2021). Enforcing the student to blindly imitate the teacher's outputs can make the student inherit such biases and produce suboptimal predictions.

To overcome this challenge, one common approach (Hinton et al., 2015) suggests scaling the teacher's logits via a temperature parameter. A proper temperature value can enhance the quality of the teacher model's output distribution by making it closer to the true label distribution (Menon et al., 2021). However, the shifting space offered by temperature scaling is limited, and the optimal temperature value relies on resource-intensive grid search. Along a separate line, label smoothing (Szegedy et al., 2016) is proposed to regularize the neural networks, and modulated loss functions (Lin et al., 2017; Leng et al., 2022) are designed to address various statistical issues in model training such as overfitting and data imbalance. Despite their potential, there is a lack of work that explores tailoring such techniques for more robust knowledge distillation.

In this study, we propose *PTLoss* for knowledge distillation, which generalizes the vanilla KL loss function and implicitly creates a debiased teacher distribution closer to the ground truth (as shown in Figure 1). Instead of forcing an out-and-out imitation of the original teacher model, PTLoss moderates the distillation objective by adding perturbations to the standard KL loss. Specifically, we first represent the KL loss using a Maclaurin series and then perturb its leading-order terms to construct a more flexible learning objective. This manipulation enables consequential adjustments to the teacher's output distribution. To determine the perturbation extent, we compute the equivalent

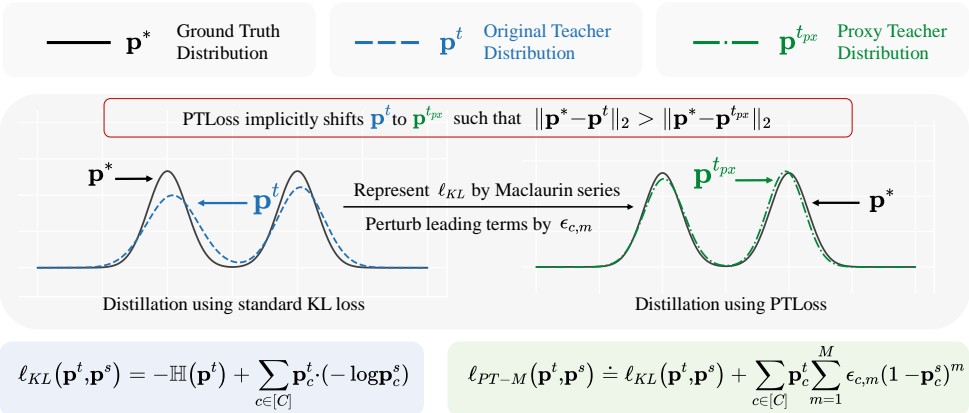

Figure 1: PTLoss implicitly transforms the original teacher into a proxy teacher with a distribution closer to the ground truth distribution. This approach addresses the issue of sub-optimal student models resulting from discrepancies between the teacher's output distribution and the ground truth distribution. By introducing perturbation to standard KL loss represented by its Maclaurin series, we obtain a better proxy teacher, which leads to a more effectively distilled student.

distribution of this implicitly shifted teacher's output distribution after perturbations (named "*proxy teacher*") and measure the empirical deviation between the proxy teacher and the ground truth data. It leads to a systematic searching strategy for the perturbation coefficients — the near-optimal perturbation coefficients should minimize the deviation between the distillation risk and the population risk on the validation set.

Theoretically, we justify the effectiveness of PTLoss by proving that it can reduce the deviation from the distillation risk compared to KL loss. We draw a connection between the PTLoss and other perturbation methods (*e.g.,* temperature scaling (Hinton et al., 2015), label smoothing (Szegedy et al., 2016), and focal loss (Lin et al., 2017)). We illustrate that the PTLoss can debias the teacher to produce higher-fidelity outputs via a finer-grained perturbation, while subsuming existing perturbation techniques as special cases. Experiments on multiple datasets with different-sized teacher models demonstrate the empirical advantages of the PTLoss.

**Contributions.** In summary, we make the following contributions: (1) A new knowledge distillation loss function PTLoss, which formulates the vanilla KD loss in the form of Maclaurin series and perturbs it to improve the fidelity of teacher models; (2) A principled method to compute the proxy teacher for determining the perturbation coefficients in PTLoss; (3) Theoretical analysis on why PTLoss can lower the distillation risk bound; and (4) Comprehensive experiments on multiple NLP datasets with different-sized teacher models showing the advantage of PTLoss.

## 2  PRELIMINARIES

**Multi-class Classification.** In a multi-class classification problem with $C$ classes, we are given a set of training examples $\mathcal{D} = \{(x_n, y_n)\}_{n=1}^N$ where input $x_n \in X$ and output $y_n$ is a one-hot vector in $Y = \{y | y \in \{0,1\}^C, \mathbf{1}^T y = 1\}$ indicating the target label of example $x_n$. The goal is to learn a probability predictor $\mathbf{p} : X \to \mathbb{R}^C$ by optimizing the below minimal *risk*:

$$R(\mathbf{p}) = \mathbb{E}_{(x,y)}[\ell(y, \mathbf{p}(x))]. \tag{1}$$

where $\ell(y, \mathbf{p}(x))$ is the loss of predicting $\mathbf{p}(x)$ when the true label of example $x$ is $y$.

A canonical loss function is the cross-entropy loss: $\ell_{CE}(y, \mathbf{p}(x)) = -y \log(\mathbf{p}(x))$ and we may further approximate the above risk via the *empirical risk* on the training set $\mathcal{D}$:

$$\hat{R}(\mathbf{p}; \mathcal{D}) \doteq \frac{1}{N} \sum_{n=1}^N y_n(-\log(\mathbf{p}(x_n))) \tag{2}$$

**Our Problem Formulation.** In this work, we study the knowledge distillation problem where the labeled training set $\mathcal{D}$ is inaccessible[1]. Specifically, we are only given an unlabeled distillation set $\mathcal{D}_u$, a teacher model $\mathbf{p}^t$, and asked to learn a student model $\mathbf{p}^s$.

**Standard Distillation Strategy.** A standard KD strategy Hinton et al. (2015) is to replace the ground truth one-hot label $y_n$ in Eq. 2 with the teacher model's output probabilistic label estimate $\mathbf{p}^t(x_n)$ and utilize the KL divergence loss to learn the student model $\mathbf{p}^s$ via the *distillation empirical risk*:

$$\tilde{R}_{KL}(\mathbf{p}^s; \mathbf{p}^t, \mathcal{D}_u) \doteq \frac{1}{N_u} \sum_{n=1}^{N_u} \ell_{KL}\left(\mathbf{p}^t(x_n), \mathbf{p}^s(x_n)\right), \tag{3}$$

where $N_u = |\mathcal{D}_u|$ and $\ell_{KL}(\mathbf{p}, \mathbf{q}) = KL(\mathbf{p}\|\mathbf{q}) = \mathbf{p}^T \log(\mathbf{p}) - \mathbf{p}^T \log(\mathbf{q})$.

## 3 PERTURBED DISTILLATION LOSS

Using the KL divergence loss (in short "KL loss") for distillation essentially assumes the teacher model is perfect and forces the student model to mimic the teacher's output label distribution. In reality, the teacher model can produce a biased estimate of label distribution and lead to a sub-optimal student model, as demonstrated by both theoretical analysis (Menon et al., 2021) and empirical observations (Müller et al., 2019) (as well as our experiments in Section 5.1).

In this work, we present a new distillation loss that generalizes the standard KL loss to accommodate various degrees of distribution gaps between the biased teacher's output distribution and the underlying ground truth distribution. Inspired by the PolyLoss (Leng et al., 2022), we propose to first replace the logarithmic terms in the standard KL loss with their corresponding Maclaurin series and then perturb the polynomial terms as follows:

$$\log(x) = -\sum_{m=1}^{\infty} \frac{(1-x)^m}{m} \xrightarrow[\text{term coefficients}]{\text{Perturb polynomial}} \log(x) \approx -\sum_{m=1}^{\infty} (\frac{1}{m} + \epsilon_m)(1-x)^m \tag{4}$$

Here, we essentially replace the original coefficient $\frac{1}{m}$ of the $m$-th order polynomial term in the standard KL loss to $(\frac{1}{m} + \epsilon_m)$. By further replacing the logarithmic terms in standard KL loss (Eq. 3) with the above Eq. 4, we will have:

$$\ell_{KL}\left(\mathbf{p}^t(x_n), \mathbf{p}^s(x_n)\right) = -\mathbb{H}\left(\mathbf{p}^t(x_n)\right) + \sum_{c \in [C]} \mathbf{p}_c^t(x_n)[-\log \mathbf{p}_c^s(x_n)]$$

$$\approx -\mathbb{H}\left(\mathbf{p}^t(x_n)\right) + \sum_{c \in [C]} \mathbf{p}_c^t(x_n)\left[-\log \mathbf{p}_c^s(x_n) + \sum_{m=1}^{\infty} \epsilon_{c,m}(1 - \mathbf{p}_c^s(x_n))^m\right], \tag{5}$$

where $\mathbf{p}_c^t(x_n)$ and $\mathbf{p}_c^s(x_n)$ denote the probability that example $x_n$ belongs to the class $c$ according to the teacher (student) model, and $\mathbb{H}\left(\mathbf{p}^t(x_n)\right)$ is the entropy of the teacher output distribution.

We can further separate out the perturbation coefficients on the right hand side of Eq. 5 and merge $\sum_{c \in [C]} \mathbf{p}_c^t(x_n)\left[-\log \mathbf{p}_c^s(x_n)\right]$ with $\mathbb{H}\left(\mathbf{p}^t(x_n)\right)$ to obtain our perturbed distillation loss:

$$\ell_{PT}\left(\mathbf{p}^t(x_n), \mathbf{p}^s(x_n)\right) \doteq \ell_{KL}\left(\mathbf{p}^t(x_n), \mathbf{p}^s(x_n)\right) + \sum_{c \in [C]} \mathbf{p}_c^t(x_n) \sum_{m=1}^{\infty} \epsilon_{c,m}(1 - \mathbf{p}_c^s(x_n))^m. \tag{6}$$

The above equation presents our perturbed distillation loss in its most general form. In practice, however, we cannot tune infinite number of coefficients $\epsilon_{c,m}$ and thus we propose to only tune the first $M$ leading polynomial coefficients while keeping the rest unchanged as follows:

$$\ell_{PT\text{-}M}\left(\mathbf{p}^t(x_n), \mathbf{p}^s(x_n)\right) \doteq \ell_{KL}\left(\mathbf{p}^t(x_n), \mathbf{p}^s(x_n)\right) + \sum_{c \in [C]} \mathbf{p}_c^t(x_n) \sum_{m=1}^{M} \epsilon_{c,m}(1 - \mathbf{p}_c^s(x_n))^m. \tag{7}$$

We can see that if we set all $\epsilon_{c,m}$ to 0, the $\ell_{PT}$ falls back to the $\ell_{KL}$ and thus the perturbed distillation loss can be considered as a generalization of the standard KL loss.

---

[1]This setting reflects the real-world scenario where large teacher models (e.g., ChatGPT OpenAI (2022) and GPT4 OpenAI (2023)) only expose their outputs and/or APIs without original training data because of their large model sizes and cautions toward data leakage/misuse.

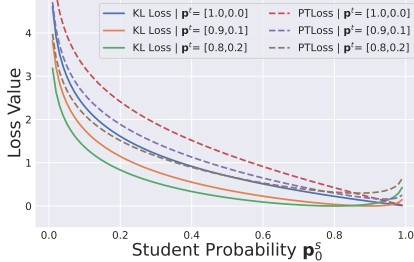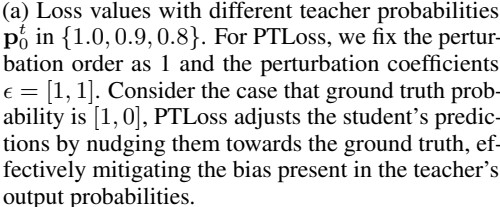

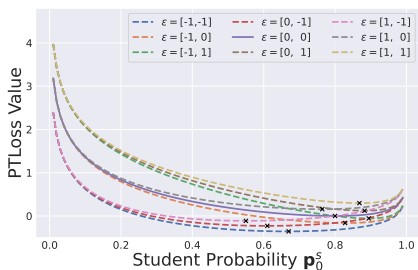

(a) Loss values with different teacher probabilities $\mathbf{p}_0^t$ in $\{1.0, 0.9, 0.8\}$. For PTLoss, we fix the perturbation order as 1 and the perturbation coefficients $\epsilon = [1, 1]$. Consider the case that ground truth probability is $[1, 0]$, PTLoss adjusts the student's predictions by nudging them towards the ground truth, effectively mitigating the bias present in the teacher's output probabilities.

(b) PTLoss values with different perturbations. We fix the teacher probability $\mathbf{p}_0^t = [0.8, 0.2]$ and vary the perturbation coefficients $\epsilon$, while the perturbation order is always fixed to 1. The black cross denotes the best student model output $\mathbf{p}_0^s$ that achieves the lowest loss value. This shows PTLoss can enable flexible adjustments to the loss curve and effectively reduces the bias of the teacher's output.

Figure 2: Intuitive understanding of PTLoss in binary classification.

Figure 2 presents how PTLoss adjusts biased teachers. For visualization simplicity, we set the number of classes $C = 2$. In Figure 2a, we vary the teacher probability to show how the biased teacher model will impact the distilled student model under either the standard KL loss or our proposed PTLoss. We observe that PTLoss can guide the student's predictions toward the ground truth and thus effectively reduces the inherent bias in the teacher's output probabilities. In Figure 2b, we demonstrate PTLoss enables a diverse shift space to the loss curve. By setting the perturbation coefficients, PTLoss allows flexible adjustments to the loss curve. Combining with our perturbation coefficients selection methods discussed in Sec. 4.3, we can determine the perturbation to optimize the distillation process.

**Connections to other perturbation methods**. In Appendix A.1, we establish the connections between PTLoss and other related methods that transform the teacher output probabilities, such as label smoothing Szegedy et al. (2016), temperature scaling Hinton et al. (2015), and focal loss Lin et al. (2017). We show that the loss shift space produced by PTLoss encompasses these alternative techniques and thus PTLoss can offer additional adjustment capabilities.

## 4 PROXY TEACHER AND THE PRINCIPLE OF SELECTING POLYNOMIAL COEFFICIENTS

In this section, we first present a theorem to show how the teacher model affects the gap of a student model's distillation empirical risk and its population risk (§ 4.1). Then, we demonstrate that using PTLoss implicitly transforms the original teacher model to a *proxy teacher* under the KL loss. Based on the above theorem, we know when this proxy teacher distribution is closer to the true distribution, we will have a better distilled student model (§ 4.2). Finally, we establish our principle of selecting the perturbation coefficients in PTLoss: searching the coefficients that lead to a proxy teacher closest to the empirical estimate of true distribution on a validation set (§ 4.3).

### 4.1 THE CONNECTION OF THE TEACHER MODEL AND THE RISKS OF STUDENT MODEL

**Theorem 1.** *Given a teacher model $\mathbf{p}^t$, an unlabeled distillation dataset $\mathcal{D}_u$ with an unknown true distribution $\mathbf{p}^*$, we have for any probability predictor $\mathbf{p} : \mathcal{X} \to \mathbb{R}^C$:*

$$\mathbb{E}\left[(\tilde{R}_{KL}(\mathbf{p}; \mathbf{p}^t, \mathcal{D}_u) - R(\mathbf{p}))^2\right] \leq \frac{2}{N_u} \cdot \mathbb{V}\left[\mathbf{p}^t(x)^T \log(\mathbf{p}(x))\right] +$$
$$\mathcal{O}\left(\left(\mathbb{E}_x[\|\mathbf{p}^t(x) - \mathbf{p}^*(x)\|_2]\right)^2 + \mathbb{E}_x\left[\left(\mathbf{p}^t(x)^T \log \mathbf{p}^t(x)\right)^2\right]\right),$$

*where $\mathbb{V}[\cdot]$ denotes the variance of a random variable.*

We defer the detailed proofs of above theorem to Appendix A.2 and focus on its implications here. We can see that the gap between a model $\mathbf{p}$'s distillation empirical risk and its population risk depends

on three terms: (1) the variance of its KL distance to the teacher model $\mathbf{p}^t$, (2) the $L_2$ distance between the teacher model output distribution $\mathbf{p}^t$ and the true distribution $\mathbf{p}^*$, and (3) the entropy of the teacher distribution. In practice, obtaining a sizable unlabeled distillation set $\mathcal{D}_u$ is relatively straightforward, which leads to a large value of $N_u$. As a result, the first term (of order $O(1/N_u)$) will converge to 0 as $N_u$ keeps increasing and the latter two terms (one quantifies the distance between teacher $\mathbf{p}^t$ and true $\mathbf{p}^*$, and the other quantifies the teacher's uncertainty) will dominate the risk gap. This observation also resonates with our intuition that an accurate and well-calibrated teacher yields better improved bounds on the generalization error of the student.

## 4.2 THE EQUIVALENCE OF PROXY TEACHER UNDER KL LOSS AND ORIGINAL TEACHER UNDER PTLOSS

The above theorem states that an ideal teacher model, when used in KL loss for distillation, should output a distribution as close to the true distribution as possible. In reality, however, the teacher model is usually fixed. Here, we show that using PTLoss for distillation can implicitly transform the original teacher to a *proxy teacher* under the KL loss. Namely, given the original teacher model $\mathbf{p}^t$ and a set of perturbation coefficients $\{\epsilon_{c,m}\}$ in PTLoss, we can obtain a proxy teacher $\mathbf{p}^{t_{px}}$ such that:

$$
\tilde{R}_{KL}(\mathbf{p}^s; \mathbf{p}^{t_{px}}, \mathcal{D}_u) = \tilde{R}_{PT\text{-}M}(\mathbf{p}^s; \mathbf{p}^t, \mathcal{D}_u)
$$
$$
= \frac{1}{N_u} \sum_{n=1}^{N_u} \ell_{PT\text{-}M}(\mathbf{p}^t(x_n), \mathbf{p}^s(x_n)),
$$
(8)

which establishes the equivalence of proxy teacher under KL loss and original teacher under PTLoss. With the proxy teacher $\mathbf{p}^{t_{px}}$, we aim to determine the best perturbation coefficients $\{\epsilon_{c,m}\}$. Note for each $\{\epsilon_{c,m}\}$, we can obtain a proxy teacher. We illustrate how we obtain the proxy teacher in the rest of this subsection, and discuss how to select the best perturbation coefficients in § 4.3.

Intuitively, the proxy teacher is derived by solving the below optimization problem:

$$
\min_{\mathbf{p}^{t_{px}}} \quad \|\tilde{R}_{PT\text{-}M}(\mathbf{p}^s; \mathbf{p}^t, \mathcal{D}_u) - \tilde{R}_{KL}(\mathbf{p}^s; \mathbf{p}^{t_{px}}, \mathcal{D}_u)\|_2.
$$
(9)

In practice, however, we do not need the above risk equivalence in Eq. 8 to hold for all possible student models $\mathbf{p}^s$. Instead, we focus on the minimizer of the left-hand side of Eq. 8 because it is practically close to the final learned student model. By substituting this minimizer $\mathbf{p}^s = \mathbf{p}^{t_{px}}$ into Eq. 9, the second term in the norm of Eq. 9 becomes 0, and the first term could be expanded by its definition in Eq. 8, we thus have the following objective:

$$
\min_{\mathbf{p}^{t_{px}}} \quad \left\| \frac{1}{N_u} \sum_{n=1}^{N_u} \left( \ell_{KL}(\mathbf{p}^t(x_n), \underline{\mathbf{p}^{t_{px}}(x_n)}) + \sum_{c \in [C]} \mathbf{p}_c^t(x_n) \sum_{m=1}^{M} \epsilon_{c,m}(1 - \underline{\mathbf{p}_c^{t_{px}}(x_n)})^m \right) \right\|_2.
$$
(10)

This objective enables us to solve $\mathbf{p}^{t_{px}}$ given $\mathbf{p}^t$ and $\{\epsilon_{c,m}\}$, where $\mathbf{p}^t$ is the teacher's output probability on the validation set, and $\{\epsilon_{c,m}\}$ is a given set of perturbation coefficients. However, this optimization problem is nonlinear and lacks a closed-form analytical solution. Consequently, we compute the $\mathbf{p}^{t_{px}}$ using the numerical approach[2] and the details are discussed in Appendix A.3. We have also considered an alternative solution to this optimization problem, which involves defining a parameterized function $g_\theta(\cdot) : [0, 1]^C \rightarrow [0, 1]^C$ that explicitly transforms the original teacher to the proxy teacher, namely $g_\theta(\mathbf{p}^t) = \mathbf{p}^{t_{px}}$. We would then find the best $\theta$ minimizing the above objective (possibly via gradient-based methods). This approach leads to a smooth proxy teacher but also introduces bias from the function class defined by $\theta$. Therefore, we leave it to future work and resort to the numerical approach in this study.

## 4.3 SELECTING PERTURBATION COEFFICIENTS VIA THE BEST PROXY TEACHER

For each candidate set of perturbation coefficients $\{\epsilon_{c,m}\}$ in PTLoss, we can find a corresponding proxy teacher and compute its risk deviation upper bound according to theorem 1. In practice, the size

---

[2] We use a hybrid algorithm of the Newton-Raphson method and the Levenberg-Marquardt algorithm as defined in 'scipy.optimize.fsolve' https://docs.scipy.org/doc/scipy/reference/generated/scipy.optimize.fsolve.html.

---

**Algorithm 1** Automated Perturbation Coefficients Selection

---

**Require:** Validation set $\mathcal{D}_v$, Teacher model $\mathbf{p}^t$, Max perturbation order $N_M$, Max search trails $N_k$, Perturbation coefficient search space $\mathcal{S}$.

  Initialize $\{\epsilon^*_{c,m}\} \leftarrow \{\}$, $\hat{Q}^* \leftarrow \infty$.
  **for** $M = 1$ to $N_M$ **do**
    **for** $k = 1$ to $N_k$ **do**
      Randomly sample a set of perturbation coefficients $\{\epsilon_{c,m}\}$ from $\mathcal{S}$.
      Solve the proxy teacher $\mathbf{p}^{t_{px}}$ given $\{\epsilon_{c,m}\}$ and $\mathbf{p}^t$ via Eq. 10.
      Compute the quality score of perturbation coefficients $\hat{Q}(\{\epsilon_{c,m}\})$ via Eq. 11.
      **if** $\hat{Q}(\{\epsilon_{c,m}\}) < \hat{Q}^*$ **then**
        $\hat{Q}^* \leftarrow \hat{Q}(\{\epsilon_{c,m}\})$, $\{\epsilon^*_{c,m}\} = \{\epsilon_{c,m}\}$.
      **end if**
    **end for**
  **end for**
  **return** $\{\epsilon^*_{c,m}\}$.

---

of distillation set $N_u$ is typically large and thus we can omit the $O(1/N_u)$ variance term. Furthermore, since the ground truth distribution $\mathbf{p}^*$ is unknown, we use an unbiased estimator to replace it. Finally, we replace the expectation by the sample mean and define the empirical risk below:

$$\hat{Q}(\{\epsilon_{c,m}\}) = \left(\frac{1}{N_v}\sum_{n=1}^{N_v}\left[\|\mathbf{p}^{t_{px}}(x_n) - \mathbf{y}_n\|_2\right]\right)^2 + \frac{1}{N_v}\sum_{n=1}^{N_v}\left[\left(\mathbf{p}^{t_{px}}(x_n)^T\log\mathbf{p}^{t_{px}}(x_n)\right)^2\right], \quad (11)$$

where $N_v$ is the size of validation set and $\mathbf{y}_n$ is a one-hot label vector of $x_n$, serving as the unbiased estimation of $\mathbf{p}^*(x_n)$. We use $\hat{Q}(\{\epsilon_{c,m}\})$ as a "quality score" for each candidate coefficients set. Users can define a search space of $\{\epsilon_{c,m}\}$ and we will pick the optimal $\{\epsilon^*_{c,m}\}$ that minimizes $\hat{Q}$. We present the pseudo-code for selecting perturbation coefficients in Algorithm 1, and the search time for perturbation coefficients is detailed in Appendix A.4.

## 5 EXPERIMENTS

In this section, we first conduct experiments on a synthetic dataset to verify our assumption that the teacher outputting a distribution closer to the ground truth distribution leads to a better student (§5.1). Then, we present our main results on 6 real-world NLP datasets (§5.2). Moreover, we show how the proxy teacher enhances the distillation process(§5.3). In the appendix, we further evaluate the performance of PTLoss on CIFAR-100 to show its potential in computer vision tasks (§A.8).

### 5.1 EXPERIMENTS ON SYNTHETIC GAUSSIAN DATASET

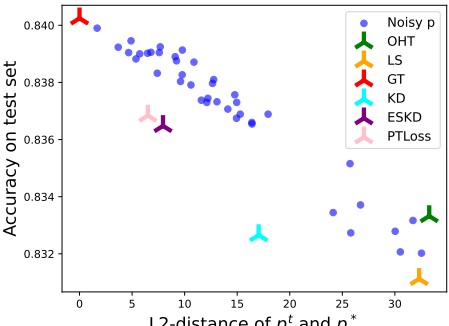
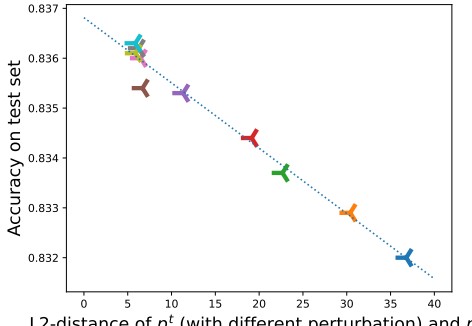

(a) OHT - one-hot training; LS - label smoothing; GT - ground truth; KD - knowledge distillation; ESKD: early-stopped KD (Ren et al., 2022); PTLoss: here we use 3-order perturbation.

(b) Correlation between the $L_2$-distance (between the teacher model $\mathbf{p}^t$ with different levels of perturbations and the ground truth $\mathbf{p}^*$) and the test accuracy of the student model.

Figure 3: Experiments on a synthetic Gaussian dataset.

We first conduct an illustrative experiment with a synthetic dataset where the ground truth distribution $\mathbf{p}^*(x)$ is known. Specifically, we follow (Ren et al., 2022) to generate $10^5$ examples from a mixture of Gaussian distribution and train an MLP with 3 hidden layers on this synthetic dataset[3]. We compare PTLoss with 4 baselines: one-hot supervision (OHT), label smoothing (LS), standard knowledge distillation (KD), and early-stopped knowledge distillation (ESKD) (see details in below § 5.2). As illustrated in Fig. 3a, the quality of the distilled student improves as the $L_2$-distance between the teacher distribution and the ground truth distribution decreases. On this synthetic Gaussian dataset, PTLoss also outperforms the baselines after adding a 3-order perturbation.

In Fig. 3b, we sample 10 proxy teachers in different stages of the perturbation coefficient searching process (§4.3) and compare their results. It is clear that a teacher model with a smaller $L_2$-distance to the ground truth distribution can lead to a better student model. This observation verifies our hypothesis in Eq. 11 — searching a proxy teacher closer to the ground truth distribution can reduce the empirical deviation and improve the distilled student model.

## 5.2 EXPERIMENTS ON NATURAL LANGUAGE DATASETS

**Tasks and Datasets.** We conduct our main experiments on 6 natural language datasets, including (1) CoLA (Warstadt et al., 2018) for linguistic acceptability, (2) MNLI (Williams et al., 2017) for multi-genre natural language inference, (3) MRPC (Dolan and Brockett, 2005) for paraphrase similarity matching, (4) RTE (Wang et al., 2018) for textual entailment inference, (5) SST-2 (Wang et al., 2018) for sentiment analysis, and (6) BoolQ (Clark et al., 2019) for boolean question answering. We list the detailed dataset statistics in the Appendix A.6.

**Model Architectures.** For the teacher model, we choose the T5 architecture (Raffel et al., 2020) and select two teacher models of different scales. Specifically, we use T5-xxl with 11 billion parameters and T5-large with 770 million parameters. For the student model, we use the BERT-base model (Devlin et al., 2018) with 110 million parameters.

| Method | CoLA (Matt.) | MNLI (Acc.) | MRPC (F1) | RTE (Acc.) | SST-2 (Acc.) | BoolQ (Acc.) | Average |
|---|---|---|---|---|---|---|---|
| Teacher T5-xxl (#Param. = 11B) | 71.5 | 94.7 | 92.4 | 92.2 | 96.4 | 89.1 | 89.4 |
| Standard KL | 58.8 | 90.3 | 88.3 | 78.1 | 89.2 | 69.5 | 79.0 |
| Temp. Scaling (Hinton et al., 2015) | 59.4 | 90.7 | 88.9 | 79.6 | 89.4 | _72.0_ | 80.0 |
| Label Smoothing (Szegedy et al., 2016) | 59.3 | 90.6 | 89.2 | 79.2 | 89.9 | 68.6 | 79.6 |
| Focal (Lin et al., 2017) | 59.2 | 90.7 | 88.7 | 80.4 | 89.3 | 68.2 | 79.6 |
| Flooding (Ishida et al., 2020) | 58.9 | 90.6 | 89.6 | 80.2 | 89.3 | 69.3 | 79.7 |
| CRD (Tian et al., 2019) | 59.5 | 90.5 | 90.6 | 81.1 | _89.6_ | 71.8 | 80.5 |
| Annealing KD (Jafari et al., 2021) | 59.8 | 90.7 | 90.0 | 80.7 | 89.3 | 70.7 | 80.2 |
| FilterKD (Ren et al., 2022) | 59.2 | 90.7 | 89.5 | 80.4 | 89.3 | 69.6 | 79.8 |
| MetaDistill (Zhou et al., 2022) | _60.4_ | _90.8_ | **91.4** | _81.3_ | 89.5 | 71.9 | _80.9_ |
| **PTLoss (ours)** | **61.2** | **91.1** | _91.2_ | **83.5** | **90.3** | **73.1** | **81.8** |
| Teacher T5-large (#Param. = 770M) | 61.4 | 93.6 | 92.1 | 87.2 | 95.5 | 77.9 | 84.6 |
| Standard KL | 54.8 | 90.0 | 87.8 | 77.6 | 88.8 | 69.5 | 78.1 |
| Temp. Scaling (Hinton et al., 2015) | 55.6 | 90.4 | 88.7 | 79.4 | 89.2 | 70.4 | 79.1 |
| Label Smoothing (Szegedy et al., 2016) | 56.4 | 90.6 | 89.2 | 79.2 | 89.2 | 69.1 | 79.1 |
| Focal (Lin et al., 2017) | 56.0 | 90.3 | 88.4 | 79.9 | 89.3 | 68.9 | 78.8 |
| Flooding (Ishida et al., 2020) | 57.8 | 90.0 | 89.5 | 79.5 | 89.0 | 68.9 | 79.3 |
| CRD (Tian et al., 2019) | 58.2 | 90.2 | _89.8_ | 80.3 | _89.4_ | _70.5_ | 79.7 |
| Annealing KD (Jafari et al., 2021) | 58.3 | 90.4 | _89.8_ | 79.9 | _89.4_ | 69.7 | 79.6 |
| FilterKD (Ren et al., 2022) | 56.7 | 90.2 | 89.1 | 78.8 | 89.2 | 69.2 | 78.9 |
| MetaDistill (Zhou et al., 2022) | _58.6_ | **90.7** | 89.6 | _81.0_ | 89.3 | 70.4 | _80.1_ |
| **PTLoss (ours)** | **60.5** | **90.7** | **91.1** | **82.7** | **90.0** | **71.0** | **81.0** |

Table 1: Main results on natural language datasets. The student model (BERT-base) is distilled from teacher models of different sizes (T5-xxl and T5-large). All results are averaged over three runs. The bolded numbers indicate the best results, while the underscore "_" denotes the second-best results.

---

[3]See more details in Appendix A.5.

**Compared Methods.** We compare PTLoss with the following baselines: (1) Standard KL loss (Kullback, 1959): adopts standard KL divergence loss for knowledge distillation; (2) Temperature scaling (Hinton et al., 2015): scales the teacher output logits via a temperature hyper-parameter; (3) Label smoothing (Szegedy et al., 2016): smooths the teacher output class probabilities by a small scalar; (4) Focal loss (Lin et al., 2017): modulates the cross-entropy loss to focus on hard examples; (5) Flooding (Ishida et al., 2020): a regularization method to intentionally prevent further reduction of the training loss; (6) CRD (Tian et al., 2019): uses a contrastive objective in knowledge distillation; (7) AnnealingKD (Jafari et al., 2021): feeds the rich information provided by the teacher's soft-targets incrementally; (8) FilterKD (Ren et al., 2022): trains the student from the smoothed predictions of the teacher network; (9) MetaDistill (Zhou et al., 2022): evolves the teacher network with the feedback from the distilled student in a meta learning framework.

For all the baselines, we conduct an exhaustive hyper-parameter search on the validation set. For our own PTLoss method, we set its perturbation order $M = 5$ and use the proxy teacher-based method to search its perturbation coefficients (§4.3). See Appendix A.7 for more details. We run each method with three different random seeds and report its average performance.

**Main Results.** Table 1 shows the main results on 6 NLP datasets. We have the following observations: (1) PTLoss outperforms on 11 out of the total 12 tasks, achieving the best average performance across the board. The only exception is MetaDistill, which tops the results on the MRPC when using the T5-xxl teacher and ties with PTLoss on MNLI when using the T5-large teacher. (2) The advantages offered by PTLoss are robust, regardless of the scale of the teacher model. Notably, as the disparity in scale between the teacher model and student model reduces, the performance gap between them also narrows. (3) In comparison to vanilla KD, which utilizes the standard KL, PTLoss showcases significant enhancement. Specifically, it exceeds standard KL by an average of $2.8\%$ and $2.9\%$, respectively. (4) Surveying the baseline methods, MetaDistill stands out, securing the second-highest performance across most tasks. On the whole, the cluster of KD methods generally outstrips the simple regularization methods.

## 5.3 PROXY TEACHER ANALYSIS

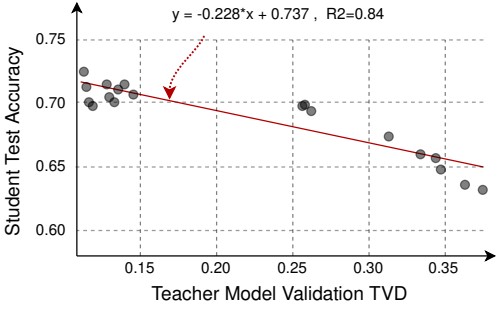 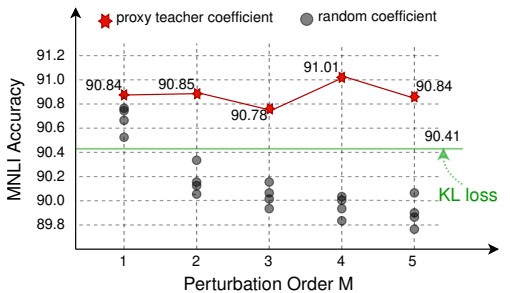

(a) Correlation between the validation TVD of the teacher model and the test accuracy of the student model. Experiments are conducted on BoolQ.

(b) The Proxy Teacher method v.s. random search for the perturbation coefficient selection. We conduct experiments on MNLI with a T5-xl teacher.

Figure 4: PTLoss analysis.

**Correlation between teacher's distance to ground truth and student's performance.** To explore where PTLoss's performance gains come from, we train multiple teacher models on the BoolQ dataset and distill them into the student models. Fig. 4a shows that the student model performance on the test set is highly correlated with the total variance distance (TVD) (*i.e.*, $l_\infty$ distance) between the teacher model's output distribution and the ground truth distribution on the validation set. This result resonates with our findings from synthetic datasets(§ 5.1), confirming that, on real-world datasets, a teacher model with a predictive distribution closer to the ground truth can produce a more effectively distilled student.

**Effectiveness of Perturbation Coefficients Search.** We continue to validate the effectiveness of the proxy teacher-based perturbation coefficients selection method using MNLI as a representative dataset. Specifically, we vary the perturbation order $M$ from 1 to 5 and report the performance of the student models distilled via PTLoss with different perturbation coefficients. These coefficients are obtained either by minimizing the empirical risk deviation of proxy teacher (c.f. Eq. 10) or via

random sampling from the space of $[-1, 10]^M$. As shown in Fig. 4b, the coefficients obtained from our proxy teacher based method can achieve consistent improvements over the random coefficients. If we just randomly set the perturbation coefficients, the student performance can drop by up to $1.2\%$. Also, by comparing different perturbation orders, we find that the higher the perturbation order, the greater the performance differences. This is because in the higher-dimension space, it is harder for random search to get a set of appropriate perturbation coefficients, which makes the random PTLoss even worse than the standard KL loss. Conversely, equipped with the perturbation coefficients obtained via proxy teacher, PTLoss can significantly outperform the underlying KL loss.

## 6 RELATED WORK

**Knowledge Distillation.** Knowledge distillation was first proposed in (Buciluǎ et al., 2006) to compress the large models to smaller, faster models without a significant performance drop. Hinton et al. (2015) generalized this technique by introducing a temperature parameter to smooth the teacher model prediction and Tian et al. (2019) employed contrastive learning to train the student model. Later, Yuan et al. (2020) explored the connection between KD and label smoothing, Jafari et al. (2021) and Chen et al. (2021) studied the feeding mechanism of the teacher's knowledge. Zhao et al. (2022) decoupled the classical loss to target classes and non-target classes for KD efficiency and flexibility. Ren et al. (2022) investigated supervisory signals and proposed to average teacher outputs for KD stability, while Zhou et al. (2022) evolves the teacher model with the student feedback in a meta learning framework.

**Distillation Theory.** Concurrent with the empirical success of knowledge distillation, numerous works aim to understand its mechanisms. Hinton et al. (2015) suggest that teacher's soft labels offer "dark knowledge" through weights on incorrect labels. Menon et al. (2021) present a statistical view, observing that a good teacher model should be Bayesian to reduce the student objective variance. Stanton et al. (2021) highlight discrepancies between teacher and student output distributions and emphasize the optimization challenge in distillation. While more recent studies (Ji and Zhu, 2020; Zhou et al., 2021; Hsu et al., 2021; Allen-Zhu and Li, 2023) explore distillation from several various angles, a gap remains between the theoretical analysis and the improved distillation techniques.

**Loss Function Design.** Our work also relates to loss function design and learning. Lin et al. (2017) propose reshaping the cross-entropy loss to concentrate on hard examples and address the data imbalance issue. Leng et al. (2022) expand cross-entropy loss and focal loss into a linear combination of polynomial functions, primarily studying Poly-1 formulation on computer vision tasks while avoiding issues with high-order polynomial hyper-parameter searches. TaylorGLO (Gonzalez and Miikkulainen, 2021) utilizes Covariance Matrix Adaptation Evolution Strategy (CMA-ES) to optimize multivariate Taylor parameterization of a loss function and learning rate schedule, but lacks principled analysis on performance gains after perturbation. In contrast, we theoretically and empirically prove the necessity of adding perturbations to the KD learning objective when using a high-fidelity teacher for quality student supervision.

## 7 CONCLUSIONS AND FUTURE WORK

In this study, we proposed a novel knowledge distillation loss PTLoss which implicitly shifts the teacher model output distribution to a high-fidelity one for student model training. We also established connections between PTLoss and other loss functions by demonstrating that PTLoss can subsume the others while providing more flexible adjustments to teacher models. We theoretically showed how the teacher model affects the student model risks and presented a principled method to systematically search perturbation coefficients. Extensive experiments on multiple tasks verified our proposed theory and validated the effectiveness of distillation via PTLoss.

While PTLoss enables better KD by creating a proxy teacher closer to the ground truth distribution, we focus on the single-teacher-single-student setting in this work. It is worth exploring how this approach can be extended to ensemble KD involving multiple teachers or students. Additionally, although the proposed coefficients selection method provides a principal way to determine the perturbation hyperparameters, it remains challenging to scale up the number of classes and the perturbation order. Future work could benefit from developing scalable methods for hyperparameter search, enabling rapid determination of perturbation coefficients even in high-dimensional spaces with numerous classes or high perturbation orders.

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

# A APPENDIX

## A.1 CONNECTIONS BETWEEN PTLOSS AND OTHER PERTURBATION METHODS

**KL Loss.** The connection between PTLoss and the standard KL loss is quite direct. As we represent the standard KL loss in Maclaurin series and add perturbations, we can easily revert PTLoss to the standard KL loss by setting all perturbation coefficients $\epsilon_{c,m}$ in Eq. 7 to 0.

**Focal Loss.** Focal loss incorporates a factor $(1 - p)^\gamma$ in the loss function. We demonstrate PTLoss can subsume focal loss by expressing the perturbation coefficients as a function of the factor $(1 - p)^\gamma$. For simplicity, we denote $\mathbf{p}^t(x_n)$ and $\mathbf{p}^s(x_n)$ as $\mathbf{p}^t$ and $\mathbf{p}^s$ in the following derivation. First, by applying Focal loss to KD, we have:

$$\ell_{focal}\left(\mathbf{p}^t, \mathbf{p}^s\right) = -\mathbb{H}\left(\mathbf{p}^t\right) + \sum_{c \in [C]} \mathbf{p}_c^t (1 - \mathbf{p}_c^s)^\gamma [-\log \mathbf{p}_c^s], \tag{12}$$

where $(1 - \mathbf{p}_c^s)^\gamma$ is a factor and the parameter $\gamma > 0$ reduces the relative loss for well-classified examples.

To bridge the connection between PTLoss and the focal loss, we compare Eq. 12 to Eq. 5 and establish the following relationship:

$$\sum_{m=1}^{\infty} (\frac{1}{m} + \epsilon_{c,m})(1 - \mathbf{p}_c^s)^m = \sum_{m=1}^{\infty} \frac{(1 - \mathbf{p}_c^s)^\gamma}{m} \cdot (1 - \mathbf{p}_c^s)^m, \tag{13}$$

which leads to the perturbation coefficients as follows:

$$\epsilon_{c,m} = \frac{(1 - \mathbf{p}_c^s)^\gamma - 1}{m}. \tag{14}$$

By incorporating the derived perturbation coefficients $\epsilon_{c,m}$ in our proposed method, we demonstrate that PTLoss can effectively subsume the focal loss. In other words, PTLoss generalizes the focal loss and can adapt to various modulating factors $(1 - p)^\gamma$ to handle the class imbalance problem and improve knowledge distillation performance.

**Temperature Scaling.** We compare PTLoss with temperature scaling and claim that PTLoss subsumes it with appropriate approximation. As described in Hinton et al. (2015), the logits are adjusted by a temperature to control sharpness or smoothness of the probability distribution[4]:

$$\mathbf{p}_{0,\tau} = \frac{exp(z_0/\tau)}{exp(z_0/\tau) + exp(z_1/\tau)}, \tag{15}$$

where $\tau$ is the temperature and $z_c$ is the logits. Denote the probability without temperature scaling as $\mathbf{p}_0$, we have

$$\frac{\mathbf{p}_{0,\tau}}{\mathbf{p}_0} = \frac{exp(z_0/\tau)}{exp(z_0/\tau) + exp(z_1/\tau)} \times \frac{exp(z_0) + exp(z_1)}{exp(z_0)} = \frac{1 + exp(z_1 - z_0)}{1 + exp(\frac{z_1 - z_0}{\tau})}. \tag{16}$$

In practice, we have

$$\frac{1 + exp(z_1 - z_0)}{1 + exp(\frac{z_1 - z_0}{\tau})} \approx 1 \quad \text{or} \quad \frac{exp(z_1 - z_0)}{exp(\frac{z_1 - z_0}{\tau})} \tag{17}$$

because $|z_1 - z_0| \gg 0$. Then we have

$$\mathbf{p}_{0,\tau} \approx \mathbf{p}_0 \quad \text{or} \quad \frac{exp(z_1 - z_0)}{exp(\frac{z_1 - z_0}{\tau})} \mathbf{p}_0. \tag{18}$$

For the first case where $\mathbf{p}_{0,\tau} \approx \mathbf{p}_0$, we omit the discussion as it aligns with the standard KL loss. For the second case, we proceed to draw its connection with PTLoss as follows. We denote $\mathbf{p}_{\tau,c}^t, \mathbf{p}_{\tau,c}^s$ as the teacher, student probability scaled by temperature $\tau$. Incorporating temperature scaling, the KL loss can be formulated as:

$$\ell_{KL}^{temp}\left(\mathbf{p}_\tau^t, \mathbf{p}_\tau^s\right) = -\mathbb{H}\left(\mathbf{p}_\tau^t\right) + \sum_{c \in [C]} \mathbf{p}_{\tau,c}^t \cdot (-\log \mathbf{p}_{\tau,c}^s), \tag{19}$$

---

[4]We use binary classification for the derivation simplicity without loss of generality.

Substituting $\mathbf{p}_{\tau,c}^t$ and $\mathbf{p}_{\tau,c}^s$ in Eq. 19 using Eq. 18, we obtain

$$\ell_{KL}^{temp}\left(\mathbf{p}_\tau^t, \mathbf{p}_\tau^s\right) = -\mathbb{H}\left(\mathbf{p}_\tau^t\right) + \sum_{c\in[C]} a \cdot \mathbf{p}_c^t \cdot [-\log(b \cdot \mathbf{p}_c^s)], \tag{20}$$

where $a = exp(z_1^t - z_0^t)/exp(\frac{z_1^t - z_0^t}{\tau})$, and $b = exp(z_1^s - z_0^s)/exp(\frac{z_1^s - z_0^s}{\tau})$.

Comparing above Eq. 20 with Eq. 5, we can set

$$\sum_{m=1}^\infty (\frac{1}{m} + \epsilon_{c,m}) \cdot (1 - \mathbf{p}_c^s)^m = a[\sum_{m=1}^\infty \frac{1}{m} \cdot (1 - \mathbf{p}_c^s)^m + \sum_{m=1}^\infty \frac{1}{m} \cdot (1 - b)^m]. \tag{21}$$

It leads to

$$\epsilon_{c,m} = \frac{a}{m} \cdot (1 + (\frac{1-b}{1-\mathbf{p}_c^s})^m) - \frac{1}{m}, \tag{22}$$

which indicates PTLoss encompasses the temperature-scaled distillation loss. However, it is important to note that our goal is not to directly solve for the perturbation coefficients to PTLoss equivalent to temperature-scaled distillation loss. Instead, we aim to show that our approach covers the loss shift space produced by temperature scaling. As we demonstrated in Sec 4.3, we select the perturbation coefficients via the best proxy teacher.

**Label Smoothing.** We compare PTLoss with the label smoothing method and claim that label smoothing proposed in (Szegedy et al., 2016) is a special case of PTLoss. According to the implementation in Szegedy et al. (2016), we can smooth the teacher labels in KD by

$$\mathbf{p}_c^{t_{ls}} = (1 - \delta)\mathbf{p}_c^t + \delta/2, \tag{23}$$

with a smoothing parameter $\delta$. Starting from Eq. 5, we can replace the term $\mathbf{p}_c^t$ by its smooth version $\mathbf{p}_c^{t_{ls}}$. Then the original Eq. 5 with label smoothing becomes:

$$\ell_{KL}^{ls}\left(\mathbf{p}^t, \mathbf{p}^s\right) = -\mathbb{H}\left(\mathbf{p}^{t_{ls}}\right) + \sum_{c\in[C]} \mathbf{p}_c^{t_{ls}} \cdot (-\log \mathbf{p}_c^s) \tag{24}$$

For the entropy of the teacher output, the smooth version $\mathbb{H}\left(\mathbf{p}^{t_{ls}}\right)$ is different from the original $\mathbb{H}\left(\mathbf{p}^t\right)$ with only a constant $C$, which can be ignored when optimizing the loss function. We introduce $\Delta\mathbf{p}_c^t = \delta/2 - \delta\mathbf{p}_c^t$ and replace all the $\mathbf{p}_c^{t_{ls}}$ in Eq. 25 by $\mathbf{p}_c^{t_{ls}} = \mathbf{p}_c^t + \Delta\mathbf{p}_c^t$, then we get:

$$\ell_{KL}^{ls}\left(\mathbf{p}^t, \mathbf{p}^s\right) = -\mathbb{H}\left(\mathbf{p}^{t_{ls}}\right) + \sum_{c\in[C]} (\mathbf{p}_c^t + \Delta\mathbf{p}_c^t) \cdot (-\log \mathbf{p}_c^s) \tag{25}$$

Similarly, we let $\ell_{KL}^{ls}\left(\mathbf{p}^t, \mathbf{p}^s\right) = \ell_{PT}\left(\mathbf{p}^t, \mathbf{p}^s\right)$, it yields

$$\sum_{c\in[C]} (\mathbf{p}_c^t + \Delta\mathbf{p}_c^t) \sum_{m=1}^\infty \frac{1}{m} \cdot (1 - \mathbf{p}_c^s)^m = \sum_{c\in[C]} \mathbf{p}_c^t \sum_{m=1}^\infty (\frac{1}{m} + \epsilon_{c,m}) \cdot (1 - \mathbf{p}_c^s)^m. \tag{26}$$

We obtain

$$\epsilon_{c,m} = \frac{\Delta\mathbf{p}_c^t}{m\mathbf{p}_c^t} \xrightarrow{\Delta\mathbf{p}_c^t = \delta/2 - \delta\mathbf{p}_c^t} \epsilon_{c,m} = \frac{\delta}{m}(\frac{1}{2\mathbf{p}_c^t} - 1). \tag{27}$$

In summary, the connection between the two losses can be expressed through a specific $\epsilon_{c,m}$, which depends on the smoothing parameter $\delta$. This derivation highlights that PTLoss generalizes the label smoothing method and provides a more flexible framework that encompasses the effects of label smoothing.

## A.2 Proof of Theorem 1

The theorem 1 states that given a teacher model $\mathbf{p}^t$, an unlabeled distillation dataset $\mathcal{D}_u$ with an unknown true distribution $\mathbf{p}^*$, we have for any probability predictor $\mathbf{p} : \mathcal{X} \rightarrow \mathbb{R}^C$:

$$\mathbb{E}\left[(\tilde{R}_{KL}(\mathbf{p}; \mathbf{p}^t, \mathcal{D}_u) - R(\mathbf{p}))^2\right] \leq \frac{2}{N_u} \cdot \mathbb{V}\left[\mathbf{p}^t(x)^T \log(\mathbf{p}(x))\right] +$$
$$\mathcal{O}\left((\mathbb{E}_x[\|\mathbf{p}^t(x) - \mathbf{p}^*(x)\|_2])^2 + \mathbb{E}_x\left[\left(\mathbf{p}^t(x)^T \log \mathbf{p}^t(x)\right)^2\right]\right),$$

where $\mathbb{V}[\cdot]$ denotes the variance of a random variable.

*Proof.* We first rewrite the population risk $R(\mathbf{p})$ with cross-entropy loss $l_{CE}$ plugged in as follow:

$$R(\mathbf{p}) = \mathbb{E}_{(x,y)}[\ell(y, \mathbf{p}(x))] = \mathbb{E}_x[\mathbb{E}_{y|x}[\ell(y, \mathbf{p}(x))]] = \mathbb{E}_x[\mathbf{p}^*(x)^T(-\log(\mathbf{p}(x)))]. \tag{28}$$

Then, we write out the distillation empirical distillation risk defined in Eq. 3 and have:

$$\tilde{R}_{KL}(\mathbf{p}; \mathbf{p}^t, \mathcal{D}_u) - R(\mathbf{p}) = \frac{1}{N_u}\sum_{n=1}^{N_u}\mathbf{p}^t(x_n)^T(-\log(\mathbf{p}(x_n))) + \frac{1}{N_u}\sum_{n=1}^{N_u}\mathbf{p}^t(x_n)^T\log(\mathbf{p}^t(x_n)) \tag{29}$$
$$- \mathbb{E}_x[\mathbf{p}^*(x)^T(-\log(\mathbf{p}(x)))].$$

We let

$$\Delta \doteq \frac{1}{N_u}\sum_{n=1}^{N_u}\mathbf{p}^t(x_n)^T(-\log(\mathbf{p}(x_n))) - \mathbb{E}_x[\mathbf{p}^*(x)^T(-\log(\mathbf{p}(x)))],$$

and

$$H \doteq \frac{1}{N_u}\sum_{n=1}^{N_u}\mathbf{p}^t(x_n)^T\log(\mathbf{p}^t(x_n)),$$

then

$$\mathbb{E}\left[(\tilde{R}_{KL}(\mathbf{p}; \mathbf{p}^t, \mathcal{D}_u) - R(\mathbf{p}))^2\right] = \mathbb{E}\left[(\Delta + H)^2\right]$$
$$\leq 2\mathbb{E}\left[\Delta^2\right] + 2\mathbb{E}\left[H^2\right] \tag{30}$$
$$= 2\mathbb{V}\left[\Delta\right] + 2\mathbb{E}\left[\Delta\right]^2 + 2\mathbb{E}\left[H^2\right]$$

where the second line is by the inequality $(a + b)^2 \leq 2a^2 + 2b^2$ and the linearity of expectation, and the third line is by $\mathbb{E}\left[\Delta^2\right] = \mathbb{V}\left[\Delta\right] + \mathbb{E}\left[\Delta\right]^2$. Observe that

$$\mathbb{E}\left[\Delta\right] = \mathbb{E}_x\left[(\mathbf{p}^t(x) - \mathbf{p}^*(x))^T(-\log(\mathbf{p}^t(x_n)))\right]$$
$$\leq \mathbb{E}_x\left[\|\mathbf{p}^t(x) - \mathbf{p}^*(x)\|_2 \cdot \|\log(\mathbf{p}^t(x_n))\|_2\right]$$
$$\leq \mathbb{E}_x\left[\|\mathbf{p}^t(x) - \mathbf{p}^*(x)\|_2 \cdot c_1 \cdot \|\log(\mathbf{p}^t(x_n))\|_\infty\right] \tag{31}$$
$$\leq c_2\mathbb{E}_x\left[\|\mathbf{p}^t(x) - \mathbf{p}^*(x)\|_2\right],$$

where the second line is by the Cauchy-Schwartz inequality, the third line is by the equivalence of norms with a constant $c_1$, and the last line is by the boundedness of the log loss term[5].

Furthermore, we notice the $R(\mathbf{p})$ term in the above $\Delta$ is a constant and thus have:

$$\mathbb{V}\left[\Delta\right] = \mathbb{V}\left[\tilde{R}_{KL}(\mathbf{p}; \mathbf{p}^t, \mathcal{D}_u)\right] = \frac{1}{N_u} \cdot \mathbb{V}\left[\mathbf{p}^t(x)^T(-\log(\mathbf{p}(x)))\right] = \frac{1}{N_u} \cdot \mathbb{V}\left[\mathbf{p}^t(x)^T\log(\mathbf{p}(x))\right], \tag{32}$$

where the last equation comes from $\mathbb{V}[aX + b] = a^2\mathbb{V}[X]$.

Finally, we plug in Eqs (31)(32) and the definition of $H$ into Eq. (30) and complete the proof. $\square$

### A.3 SOLVING PROXY TEACHER VIA NUMERICAL METHOD

We solve the optimization problem defined in Eq. 10 via numerical method. Especially, we use the algorithm defined in 'scipy.optimize.fsolve', which is a hybrid method of the Newton-Raphson method and the Levenberg-Marquardt algorithm. For better numerical stability, we actually solve the equation in logit space (instead of the vanilla probability space) and use softmax function to map it back to the final probability. Another advantage of this approach is that we remove the probability constraint of $\mathbf{p}^{t_{px}}$. We also input the analytical form of the Jacobian of our optimization objective into the solver (via the 'fprime' parameter) and set the initial estimate of $\mathbf{p}^{t_{px}}$ to be the original teacher $\mathbf{p}^t$ (via the 'x0' parameter). For all the other parameters in 'scipy.optimize.fsolve', we use their default values.

### A.4 SEARCH TIME OF PERTURBATION COEFFICIENTS

For each perturbation order, we randomly sample 100 coefficient sets from $[-1, 10]$ and find the best set that has the lowest risk deviation gap according to Eq. 11 with 1000 validation examples. The whole process takes less than two minutes on CPU with 64G memory.

---

[5]This is a common assumption defined in previous literature such as (Boucheron et al. (2005), Theorem 4.1; Menon et al. (2021), Proposition 2) and can be achieved easily in practice with regularization techniques.

A.5 SYNTHETIC GAUSSIAN DATASET GENERATION

For the experiment in Sec. 5.1, we follow the setup as described in Ren et al. (2022). Specifically, we generate a 3-class toy Gaussian dataset with 10k data points. The dataset is divided into training, validation, and test sets with a split ratio $[0.9, 0.05, 0.05]$. The underlying model in this set of experiments is a 2-layer MLP with ReLU activation, and the hidden size is 128 for each layer. We set the learning rate as $5 \times 10^{-4}$, the batch size as 32, and the training epochs as 100.

The sampling process is implemented as follows: We first choose the label $y$ using a uniform distribution across all the 3 classes. Next, we sample $x|_{y=k} \sim \mathcal{N}\left(\mu_k, \sigma^2 I\right)$ as the input signal. Here $\sigma = 2$ and $\mu_k$ is a 30-dim vector with entries randomly selected from $\{-1, 0, 1\}$.

A.6 DATASET STATISTICS

| Dataset | Task | Train | Distillation | Dev | Test |
|---------|------|-------|--------------|-----|------|
| CoLA | Linguistic Acceptability | 8.5k | 8.5k | 3k | 1k |
| MNLI | Natural Language Inference | 58.9k | 314k | 19.6k | 9.8k |
| MRPC | Paraphrase Similarity Matching | 3.7k | 3.7k | 1k | 1.7k |
| RTE | Textual Entailment Inference | 2.5k | 2.5k | 0.8k | 3k |
| SST-2 | Sentiment Analysis | 6.7k | 53.8k | 6.7k | 872 |
| BoolQ | Boolean Question Answering | 2.5k | 5.9k | 1k | 3.2k |

Table 2: Dataset Statistics

A.7 HYPER-PARAMETERS

| Hyper-parameter | Search Range |
|-----------------|--------------|
| Learning Rate | $\{2, 3, 5\} \times 10^{-5}$ |
| Batch Size | $\{8, 16, 32, 64, 128, 256\}$ |
| Temperature $T$ | $\{0.1, 0.2, 0.5, 1.0, 2.0, 5.0, 10\}$ |
| Label Smoothing $\delta$ | $\{0.02, 0.05, 0.1, 0.15, 0.2\}$ |
| Focal Loss $\tau$ | $\{0.1, 0.2, 0.5, 1, 2.0, 5.0\}$ |
| Random PTLoss $\epsilon_{c,m}$ | $[-1, 10]$ |

Table 3: The search range of hyper-parameters.

We list the search range of hyperparamters in Table 3. The search for batch size and learning rate is applied to all the methods. And for each baseline, we search for the best baseline-specific hyper-parameters.

A.8 EXPERIMENTS ON THE CIFAR-100 DATASET

| | MobileNetV2 | ShuffleNetV2 | ResNet18 | GoogleNet | DenseNet121 | ResNeXt29 |
|---|---|---|---|---|---|---|
| Baseline | 68.38 | 70.34 | 75.87 | 78.72 | 79.04 | 81.03 |
| Tf-KD | 70.14 ± 0.08 | 71.64± 0.17 | 76.60± 0.06 | 79.62± 0.43 | 79.54± 0.16 | 80.75± 0.13 |
| PTLoss | **70.62 ± 0.16** | **71.97 ± 0.10** | **77.55 ± 0.06** | **80.22 ± 0.11** | **80.22± 0.11** | **81.83 ± 0.21** |

Table 4: Results on CIFAR-100 dataset.

To evaluate PTLoss on diverse tasks, we conduct this set of experiments on the CIFAR-100 dataset. We follow (Yuan et al., 2020) to get the baseline results on the studied networks. Then we re-implement Tf-KD in (Yuan et al., 2020) as we don't have access to the ground truth data during the distillation stage. Specifically, The Tf-KD implementation is modified from Tf-KD_self to be incorporated into our setting: we modified Eq.(7) in (Yuan et al., 2020) as $L_{self} = D_{\mathrm{KL}}(p_\tau^t, p_\tau)$, where $p_\tau$ and $p_\tau^t$ are the output probability of the student model and the pre-trained student model,

reshaped by a temperature $\tau$. For our methods, we simply add 1-order perturbation to $p_\tau^t$, the $\epsilon$ is selected from $\{0.1, 0.2, 0.5\}$. From Table 4, we observe that PTLoss can still outperform those baselines, which shows the applicability of PTLoss on computer vision tasks.

## A.9  VARIATION OF THE MAIN RESULT

| Method | CoLA (Matt.) | MNLI (Acc.) | MRPC (F1) | RTE (Acc.) | SST-2 (Acc.) | BoolQ (Acc.) | Average |
|---|---|---|---|---|---|---|---|
| Teacher T5-xxl | 71.5 | 94.7 | 92.4 | 92.2 | 96.4 | 89.1 | 89.4 |
| Standard KL | $58.8 \pm 0.3$ | $90.3 \pm 0.2$ | $88.3 \pm 0.4$ | $78.1 \pm 0.1$ | $89.2 \pm 0.3$ | $69.5 \pm 0.2$ | 79.0 |
| Temp. Scaling | $59.4 \pm 0.3$ | $90.7 \pm 0.1$ | $88.9 \pm 0.3$ | $79.6 \pm 0.3$ | $89.4 \pm 0.2$ | $\underline{72.0} \pm 0.3$ | 80.0 |
| Label Smoothing | $59.3 \pm 0.6$ | $90.6 \pm 0.4$ | $89.2 \pm 0.7$ | $79.2 \pm 0.4$ | $89.9 \pm 0.3$ | $68.6 \pm 0.4$ | 79.6 |
| Focal | $59.2 \pm 0.4$ | $90.7 \pm 0.4$ | $88.7 \pm 0.9$ | $80.4 \pm 0.4$ | $89.3 \pm 0.3$ | $68.2 \pm 1.5$ | 79.6 |
| Flooding | $58.9 \pm 0.5$ | $90.6 \pm 0.4$ | $89.6 \pm 0.6$ | $80.2 \pm 0.7$ | $89.3 \pm 0.4$ | $69.3 \pm 0.5$ | 79.7 |
| CRD | $59.5 \pm 0.5$ | $90.5 \pm 0.3$ | $90.6 \pm 0.4$ | $81.1 \pm 0.2$ | $\underline{89.6} \pm 0.3$ | $71.8 \pm 0.6$ | 80.5 |
| Annealing KD | $59.8 \pm 0.3$ | $90.7 \pm 0.3$ | $90.0 \pm 0.5$ | $80.7 \pm 0.2$ | $89.3 \pm 0.5$ | $70.7 \pm 0.5$ | 80.2 |
| FilterKD | $59.2 \pm 0.4$ | $90.7 \pm 0.2$ | $89.5 \pm 0.3$ | $80.4 \pm 0.3$ | $89.3 \pm 0.2$ | $69.6 \pm 0.9$ | 79.8 |
| MetaDistill | $\underline{60.4} \pm 0.2$ | $\underline{90.8} \pm 0.3$ | $\mathbf{91.4} \pm 0.4$ | $\underline{81.3} \pm 0.1$ | $89.5 \pm 0.2$ | $71.9 \pm 0.7$ | $\underline{80.9}$ |
| **PTLoss (ours)** | $\mathbf{61.2} \pm 0.3$ | $\mathbf{91.1} \pm 0.1$ | $\underline{91.2} \pm 0.3$ | $\mathbf{83.5} \pm 0.2$ | $\mathbf{90.3} \pm 0.1$ | $\mathbf{73.1} \pm 0.5$ | **81.8** |
| Teacher T5-large | 61.4 | 93.6 | 92.1 | 87.2 | 95.5 | 77.9 | 84.6 |
| Standard KL | $54.8 \pm 0.2$ | $90.0 \pm 0.1$ | $87.8 \pm 0.3$ | $77.6 \pm 0.2$ | $88.8 \pm 0.1$ | $69.5 \pm 0.2$ | 78.1 |
| Temp. Scaling | $55.6 \pm 0.3$ | $90.4 \pm 0.1$ | $88.7 \pm 0.2$ | $79.4 \pm 0.2$ | $89.2 \pm 0.5$ | $70.4 \pm 0.6$ | 79.1 |
| Label Smoothing | $56.4 \pm 0.4$ | $90.6 \pm 0.2$ | $89.2 \pm 0.6$ | $79.2 \pm 0.4$ | $89.2 \pm 0.4$ | $69.1 \pm 1.2$ | 79.1 |
| Focal | $56.0 \pm 0.2$ | $90.3 \pm 0.1$ | $88.4 \pm 0.5$ | $79.9 \pm 0.4$ | $89.3 \pm 0.5$ | $68.9 \pm 0.5$ | 78.8 |
| Flooding | $57.8 \pm 0.3$ | $90.0 \pm 0.6$ | $89.5 \pm 0.4$ | $79.5 \pm 0.4$ | $89.0 \pm 0.4$ | $68.9 \pm 0.6$ | 79.3 |
| CRD | $58.2 \pm 0.3$ | $90.2 \pm 0.4$ | $\underline{89.8} \pm 0.3$ | $80.3 \pm 0.2$ | $\underline{89.4} \pm 0.4$ | $\underline{70.5} \pm 0.5$ | 79.7 |
| Annealing KD | $58.3 \pm 0.2$ | $90.4 \pm 0.3$ | $\underline{89.8} \pm 0.5$ | $79.9 \pm 0.1$ | $\underline{89.4} \pm 0.4$ | $69.7 \pm 0.4$ | 79.6 |
| FilterKD | $56.7 \pm 0.3$ | $90.2 \pm 0.2$ | $89.1 \pm 0.4$ | $78.8 \pm 0.2$ | $89.2 \pm 0.3$ | $69.2 \pm 0.6$ | 78.9 |
| MetaDistill | $\underline{58.6} \pm 0.2$ | $\mathbf{90.7} \pm 0.3$ | $89.6 \pm 0.1$ | $\underline{81.0} \pm 0.1$ | $89.3 \pm 0.2$ | $70.4 \pm 0.2$ | $\underline{80.1}$ |
| **PTLoss (ours)** | $\mathbf{60.5} \pm 0.2$ | $\mathbf{90.7} \pm 0.1$ | $\mathbf{91.1} \pm 0.4$ | $\mathbf{82.7} \pm 0.1$ | $\mathbf{90.0} \pm 0.2$ | $\mathbf{71.0} \pm 0.3$ | **81.0** |

Table 5: Main results on natural language datasets. The student model (BERT-base) is distilled from teacher models of different sizes (T5-xxl and T5-large). All results are averaged over three runs. The bolded numbers indicate the best results, while the underscore "_" denotes the second-best results.

