# OpenReview forum: "Knowledge Distillation with Perturbed Loss: From a Vanilla Teacher to a Proxy Teacher"
_ICLR.cc/2024/Conference — Submitted to ICLR 2024_

### Official Review · Reviewer_QQFu · 2023-10-27

**Soundness:** 3 good
**Presentation:** 2 fair
**Contribution:** 3 good
**Rating:** 5
**Confidence:** 3

**Summary:**

The authors propose perturbing the conventional KL-divergence loss used for knowledge distillation in order to reduce the possible distributional shift between the teacher predictions and the ground-truth targets. They do so by using the Maclaurin series and perturbing the leading $M$ terms of this series and show a procedure to obtain good perturbation coefficients. They support their findings with both empirical and theoretical arguments.

**Strengths:**

- The idea of perturbing the Maclaurin series of the conventional KL divergence is interesting and simple.
- The empirical results (both simulated and real datasets) shows that their proposed loss indeed can improve performance across various datasets.
- The method is theoretically grounded, and in general quite simple.

**Weaknesses:**

- Only the case of no known ground-truth targets (i.e. unlabeled distillation data) is considered, but this is only mentioned briefly on page 3, and should be made clear much earlier. Either way, it would be relevant to compare the proposed procedure to incorporating a weighted ground-truth loss as is common in many applications, as this also shifts the matching distribution towards the ground-truth distribution.
- There is no common consensus on the aim of distillation procedures; whether it is to match the teacher as well as possible (see e.g. [1]) or to get the best-performing student on the ground-truth data (see e.g. [2]). The success of self-distillation is due to not perfectly matching the teacher, and such nuance should be evident from your introduction. Currently, it merely states that matching an imperfect teacher is suboptimal. Also, what is your definition of bias here?
- You argue in the introduction that a grid search for the temperature is computationally expensive, but your proposed method still requires a search over the perturbation coefficients. Also, how does the search-time scale with $M$, number of sample perturbations, and validation set size?
- Figure 4.b.: What is the point of this experiment? Injecting perturbations and choosing such perturbations randomly is very unlikely to cause an improvement in the performance. Your method is indeed outperforming random selection, but it also should?
- Missing references to theoretical distillation works: Phuong and Lampert "Towards Understanding Knowledge Distillation", Mobahi et al. "Self-Distillation Amplifies Regularization in Hilbert Space", and Borup and Andersen "Even your Teacher Needs Guidance: Ground-Truth Targets Dampen Regularization Imposed by Self-Distillation". Also, comparison to the continuous categorical distribution would be highly relevant; Gordon-Rodriguez et al. "The continuous categorical: a novel simplex-valued exponential family".

Minor:
- Throughout the paper (especially Sections 2 and 3) you have inconsistent use of transposes of the various vectors. E.g. should $y_n$ in (2) be transposed?
- "This observation also resonates with our intuition that an accurate, well-calibrated, and certain teacher [...]". How is well-calibrated and certain not contradictions here?


[1] Stanton et al. "Does Knowledge Distillation Really Work?"

[2] Beyer et al. "Knowledge distillation: A good teacher is patient and consistent"

**Questions:**

- How does your proposed PTLoss compare to using e.g. the likelihood of the continuous categorical [3,4]?
- How does Theorem 1 differ from Proposition 3 and (8) in [5]? They appear equivalent.

[3] Gordon-Rodriguez et al. "Uses and Abuses of the Cross-Entropy Loss: Case Studies in Modern Deep Learning"

[4] Gordon-Rodriguez et al. "The continuous categorical: a novel simplex-valued exponential family"

[5] Menon et al. "A Statistical Perspective on Distillation"

---

> ### Author Response · Authors · 2023-11-22
>
> 1. ```Only the case of no known ground-truth targets (i.e. unlabeled distillation data) is considered, but this is only mentioned briefly on page 3, and should be made clear much earlier. Either way, it would be relevant to compare the proposed procedure to incorporating a weighted ground-truth loss as is common in many applications, as this also shifts the matching distribution towards the ground-truth distribution.```
>
> **R1**: Thank you for your valuable feedback on our paper. To emphasize the setting, we have used a distinctive paragraph in Sec. 2 Preliminaries, initiated with a bold, non-indented sentence to capture the reader's attention immediately. Furthermore, we have provided a more detailed explanation in the footnote on page 3 as to why this setting is reasonable. Under this setting, incorporating a weighted ground-truth loss is beyond our scope because in real-world scenario, the large teacher models (e.g., ChatGPT and GPT4) provide access only to their results and/or APIs rather than their initial training datasets with ground-truth targets.
>
> We admit that this could be made clear and earlier, we would like to mention it in the introduction part in the next version.
>
> 2. ```There is no common consensus on the aim of distillation procedures; whether it is to match the teacher as well as possible (see e.g. [1]) or to get the best-performing student on the ground-truth data (see e.g. [2]). The success of self-distillation is due to not perfectly matching the teacher, and such nuance should be evident from your introduction. Currently, it merely states that matching an imperfect teacher is suboptimal. Also, what is your definition of bias here?```
>
> **R2**: Our primary goal is to optimize for the best student performance on downstream tasks. The 'bias' mentioned in our work refers to the distribution discrepancy between the teacher's predictions and the ground truth. We observe (and empirically verify) this distribution gap is positively correlated with the student model downstream task performances. Thus, we design our perturbation methods to reduce this distribution gap and thus to improve the student model's performance on downstream tasks.
>
> 3. ```You argue in the introduction that a grid search for the temperature is computationally expensive, but your proposed method still requires a search over the perturbation coefficients. Also, how does the search-time scale with number of sample perturbations, and validation set size? ```
>
> **R3**: Thanks for pointing this out. Our method indeed requires a search over perturbation coefficients, but it requires significantly less computations than a grid search for the hyper-parameters (e.g., temperature). This is because a grid search over $k$ hyper-parameters with $n$ values involves multiple iterations of model training and consumes $O(T_{train} \times n^k)$ time where $T_{train}$ denotes the time of one-round model training. In comparison, our method searches the perturbation coefficients by computing a deterministic quality score via Eq.11. As Algorithm 1 shows, our search method does not involve model training, and only scales with the number of sample perturbations and validation set size by $O(N_k \times N_v)$ , where $N_k$ is the number of sample perturbations, and $N_v$ is the validation set size.
>
> As detailed in Appendix A.4, our search process is remarkably efficient; for instance, with $N_k=100$ and $N_v=1000$, the search completes in under two minutes on a standard CPU with 64G memory.
>
> We also provide the python code for this process. Please check the uploaded **.ipynb file in the Supplementary Material** for reference.

---

> ### Author Response · Authors · 2023-11-22
>
> 4. ```Figure 4.b.: What is the point of this experiment? Injecting perturbations and choosing such perturbations randomly is very unlikely to cause an improvement in the performance. Your method is indeed outperforming random selection, but it also should?```
> **R4**: The purpose of Figure 4(b) is to demonstrate it is necessary to use our proposed method to select the perturbation coefficients rather than just random selection. In this set of experiments, we maintained a constant search time of 50 across M values from 1 to 5, this consistency did not yield evidence that M>1 outperforms M=1. However, increasing the search time to 100 led to a difference, with M=5 reaching an accuracy of 91.12 compared to M=1's 90.87.
>
> We would like to provide more background about the perturbation order M to make it clear: M controls the granularity of the perturbation. A larger M means we can add perturbation to the higher-order terms in the polynomial, thus making finer-grained adjustments to the original loss function. The space of the perturbed loss function with a larger M encompasses that with a lower M. In practice, we use M=5 to ensure a reasonable level of perturbation detail without rendering the search space impractically large.
>
> 5. ```Missing references to theoretical distillation works: Phuong and Lampert "Towards Understanding Knowledge Distillation", Mobahi et al. "Self-Distillation Amplifies Regularization in Hilbert Space", and Borup and Andersen "Even your Teacher Needs Guidance: Ground-Truth Targets Dampen Regularization Imposed by Self-Distillation". Also, comparison to the continuous categorical distribution would be highly relevant; Gordon-Rodriguez et al. "The continuous categorical: a novel simplex-valued exponential family".```
> **R5**: Thank you for highlighting these references, they will be addressed in the updated version of the manuscript.
>
> 6. ```Throughout the paper (especially Sections 2 and 3) you have inconsistent use of transposes of the various vectors. E.g. should in (2) be transposed?```
> **R6**: There is no need for a transpose. $y_n$ represents a one-hot vector and $ p_{x_n}$ is the predicted probability distribution vector. The operation $y_n log(\mathbf{p}_{x_n})$ is a dot product, thus $y_n$ does not need to be transposed here.
>
> 7. ```"This observation also resonates with our intuition that an accurate, well-calibrated, and certain teacher [...]". How is well-calibrated and certain not contradictions here?```
>
> **R7**:  In this context, 'certain' refers to the teacher's reliability to accurately predict the correct class rather than assigning high probabilities. We understand the confusion this may have caused and have removed the term 'certain' in the revised draft to avoid such misunderstandings. Thank you for bringing this to our attention.

---

> ### Author Response · Authors · 2023-11-22
>
> 8. ```How does your proposed PTLoss compare to using e.g. the likelihood of the continuous categorical [3,4]?```
>
> **R8**: The setting of continuous categorical is different from ours. In [3], the study of CC–LS poses the learning objective under the background of regularization, instead of model distillation as we study in this paper. They do not have teacher models, but directly learn from the targets. In [4], though it includes a model distillation experiments on MINST, it follows the standard knowledge distillation setting as in (Hinton et.al, 2015), so it is not directly comparable to our work.
>
> However, we modify our settings to make the expected comparison practical. Specifically, we follow the setting of Table 2 in [4] and present the results as follows. We use M=3 for PTLoss in this set of experiments, and report the average performance of 5 runs. We omit the s/epoch metric for our method because we did not find the description of computing hardware in [4].
>
> | OBJECTIVE | ACCURACY | RMSE  | S/EPOCH |
> |-----------|----------|-------|---------|
> | DIRICHLET | 90.6%    | 0.041 | 0.9     |
> | SOFT XE   | 94.9%    | 0.029 | 0.8     |
> | CC        | 95.6%    | 0.024 | 2.2     |
> | HARD XE   | 93.2%    | ---   | 0.8     |
> | PTLoss    | 96.0%    | 0.022 | ---      |
>
>
> 9.```How does Theorem 1 differ from Proposition 3 and (8) in [5]? They appear equivalent. ```
>
> **R9**: Theorem 1 in our paper differs from Proposition 3 and Equation (8) in [5] primarily due to the inclusion of an entropy term, i.e., the third term in the equation of Theorem 1. The original paper does not account for this term as it assumes label entropy to be zero, which holds in their context. However, in the case of distillation, the assumption of zero label entropy does not apply, making the entropy term a component of our theorem.

---

### Official Review · Reviewer_Bme6 · 2023-10-31

**Soundness:** 3 good
**Presentation:** 2 fair
**Contribution:** 3 good
**Rating:** 6
**Confidence:** 3

**Summary:**

This paper argues that the teacher’s output distributions can be biased from the ground truth due to various factors. Instead of forcing an out-and-out imitation of the original teacher model, the proposed PTLoss moderates the distillation objective by adding perturbations to the standard KL loss.

**Strengths:**

1.	This paper reviews the KL loss in knowledge distillation and proposes a good arguments that the teacher’s output distributions can be biased from the ground truth due to various factors.
2.	The proposed PTLoss implicitly transforms the original teacher into a proxy teacher with a distribution closer to the ground truth distribution.

**Weaknesses:**

1.	More experiments details should be given instead of putting some statistical results.

**Questions:**

1.	Figure 3 (a) is shown to verify our assumption that the teacher outputting a distribution closer to the ground truth distribution leads to a better student. Why the OHT (grey) method achieves better accuracy than LS (yellow) with lager L2-distance between pt and p*
2.	How to quickly find a certain m in   ?
3.	How to make sure that the added perturbations term is beneficial for the learning of knowledge distillation？
4.	In Figure 1,   is  unknown, how to make sure that “PTLoss implicitly shift   to    such that   “?
5.	How long to select the perturbation coefficients for each training dataset?
6.	The derivation from Eq.4 to Eq.5 is poor. If Eq.11, if   is known, why do not use ground-truth directly?
Concerns:
Although the proposed coefficients selection method provides a principal way to determine the perturbation hyperparameters, it remains challenging to scale up the number of classes and the perturbation order.

---

> ### Author Response · Authors · 2023-11-22
>
> 1. ```More experiments details should be given instead of putting some statistical results.```
> **R1**: We appreciate your valuable feedback on our work. Due to the space limitations, we put some experiment details in the appendix. Specifically, **A.3** introduces how we solve the proxy teacher via numerical methods; **A.4** presents the search time of perturbation coefficients; **A.5** introduces the detailed setup of the synthetic Gaussian experiment; **A.7** lists the hyper-parameters.
>
> In addition, we also upload a **.ipynb file** to the **Supplementary Material** to show the code of the perturbation coefficients searching method. If you have specific questions / concerns, please let us know and we are willing to address them.
>
> 2. ```Figure 3 (a) is shown to verify our assumption that the teacher outputting a distribution closer to the ground truth distribution leads to a better student. Why the OHT (grey) method achieves better accuracy than LS (yellow) with lager L2-distance between pt and p*```
> **R2**: Thank you for highlighting the observations from Figure 3(a). The curve presented in this figure indeed indicates a general trend rather than a strict correlation. It's a normal phenomenon in such analyses that exceptions to the expected pattern occur. Furthermore,  the blue dots representing noisy data points reinforce the concept that we are observing a trend, emphasizing the non-deterministic nature of this correlation.
>
> 3. ```How to quickly find a certain m in ?```
> **R3**: If we understand this question correctly, it is asking how to quickly find the perturbation order M. As the hyperparameter to control the granularity of the perturbation, a larger M means we can add perturbation to the higher-order terms in the polynomial, thus making finer-grained adjustments to the original loss function. The space of the perturbed loss function with a larger M encompasses that with a lower M. In practice, we use M=5 to ensure a reasonable level of perturbation detail without rendering the search space impractically large.

---

> ### Author Response · Authors · 2023-11-22
>
> 4. ```In Figure 1, is unknown, how to make sure that “PTLoss implicitly shift to such that “?```
>
> **R4**: This question is not clear. We guess the reviewer is asking “$\mathbf{p^*}$ is unknown, how to make sure that PTLoss implicitly shifts $\mathbf{p^t}$ to $\mathbf{p}^{t_{px}}$ such that
> $ \| \mathbf{p}^* - \mathbf{p}^t \|_2 > \| \mathbf{p}^* - \mathbf{p}^{t_px} \|_2
> $”
> This is demonstrated in Sec. 4.4. Although the ground truth distribution $\mathbf{p^*}$ is unknown, we can use an unbiased estimator to replace it. We leverage a small-size validation dataset to compute thee sample mean to replace the expectation in terms of $\mathbf{p^*}$.
>
> For a detailed explanation of this methodology, we refer you to Theorem 1 and Eq. 11 in our paper, where we elaborate on this derivation.
>
> 5. ``` How long to select the perturbation coefficients for each training dataset?```
>
> **R5**: We introduce the details in Appendix A.4. The whole process takes less than two minutes on CPU with 64G memory. We also provide the python code for this process. Please see the uploaded **.ipynb file in the Supplementary Material** for reference.
>
>
> 6. ``` The derivation from Eq.4 to Eq.5 is poor. If Eq.11, if is known, why do not use ground-truth directly? Concerns: Although the proposed coefficients selection method provides a principal way to determine the perturbation hyperparameters, it remains challenging to scale up the number of classes and the perturbation order.```
>
> **R6**: The derivation from Eq. 4 to Eq. 5 is straightforward, involving a basic linear transformation. In Eq.5, the derivation from the $=$ to the $\approx$ leverages Eq.4.
>
> The second question is not clear, we think it pertains to “if $y_n$ is known, why do not use ground-truth directly”. Here $y_n$ is the label of the validation set, which is exclusively used for validation, not training. To be consistent, our approach, as well as the baseline methods, do not use validation data directly in the learning process.
>
> We acknowledge that scaling up the number of classes and the perturbation order is challenging. Nevertheless, we theoretically and empirically show that a moderate-order can already help close the distribution gap between the proxy teacher and the ground truth. In practice, the value of M depends on the application needs. We can balance the desired performance and the manageable size of the search space size for perturbation coefficients.

---

### Official Review · Reviewer_5qNN · 2023-11-02

**Soundness:** 3 good
**Presentation:** 4 excellent
**Contribution:** 2 fair
**Rating:** 5
**Confidence:** 4

**Summary:**

The authors propose a new distillation objective (PTLoss) motivated by the inherent limitation of the KL-based distillation loss by which the student learns to match the distribution of the teacher ignoring the difference between the latter and the ground truth distribution. Their objective instead adds perturbations to the standard KL loss via a Maclaurin expansion whose (leading) coefficients are set to better match the distribution differences between the teacher and the ground truth. They provide theoretical justification for the latter, draw connections between the proposed an objective and other existing approaches, and present experiments on a variety of NLP datasets.

**Strengths:**

The proposed objective (PTLoss) is conceptually simple, well motivated and the theoretical results in Section 4.1 justify the motivation. Connections between the proposed approach and existing alternatives are provided (in the Supplementary Material). Experiments are (for the most part) convincing and that showing the effectiveness of the perturbation coefficient search is particularly welcome.

**Weaknesses:**

The quality score in (11) requires a set of labels, y, for the validation set which is described in the text as an unbiased estimator for p^*. However, it is not sufficiently clear whether these are ground-truth labels or estimated somehow. For the latter, the problem is that is not described how to obtain such unbiased estimates and for the former, that the model requires ground truth labels, which is problematic because is not consistent with the problem formulation clearly stating "we are given an unlabeled distillation set", not mentioning a validation set. Moreover, if such labels are available, one could for instance optimize the student with those and regularize with the standard distillation loss.

Though equations (8) and (9) are well motivated, a more rigorous theoretical justification (via bounding) will strengthen the claims of the proposed objective, particularly in relation to the statements that the final learned student model and the second term of (9) becoming zero.

The results in Table 1 for all methods are an average over three trials, however, the variation is not presented.

From Figure 4(b) is not clear why the authors used M=5 in their main experiments. Moreover, it is not discussed in general how to select M or the reasoning for using M>1 given the results in Figure 4(b).

**Questions:**

Why using L2 in Figure 3(b) which is consistent with (11), but TVD in Figure 4(a)?

What is used as validation set in the main experiments, Dev in Table 2? Is there a relationship between the size of the validation set relative to the distillation set and performance gain?

---

> ### Author Response · Authors · 2023-11-22
>
> 1.```The quality score in (11) requires a set of labels, y, for the validation set which is described in the text as an unbiased estimator for p^*. However, it is not sufficiently clear whether these are ground-truth labels or estimated somehow. For the latter, the problem is that is not described how to obtain such unbiased estimates and for the former, that the model requires ground truth labels, which is problematic because is not consistent with the problem formulation clearly stating "we are given an unlabeled distillation set", not mentioning a validation set. Moreover, if such labels are available, one could for instance optimize the student with those and regularize with the standard distillation loss.```
>
> **R1**: Thanks for pointing this out. We do use the ground-truth labels in the validation set to calculate the quality score in eq. 11. In our problem formulation, we emphasize the 'unlabeled distillation set' to distinguish our approach from the traditional KD setting where besides an unlabeled distillation set, an additional human labeled training set is provided. The presence of a labeled validation set, while not explicitly mentioned in the problem statement, is implied and does not conflict with the use of an unlabeled distillation set.
>
> These ground-truth labels are solely utilized for validation purposes. We ensure that the student model is not directly optimized using this data, thus maintaining the integrity of the problem formulation. Such an implementation is consistent across all baselines employed in our experiments, providing an equitable basis for comparison.
>
>
> 2. ```Though equations (8) and (9) are well motivated, a more rigorous theoretical justification (via bounding) will strengthen the claims of the proposed objective, particularly in relation to the statements that the final learned student model and the second term of (9) becoming zero.```
>
> **R2**: Thank you for your insightful suggestions regarding the need for more rigorous theoretical bounds. Currently, our approximation-based approach works well in practice. To elaborate, on the LHS of Eq.8, what we care about is $\mathbf{p}^s = \mathbf{p}^{t_{px}}$, which is the optimal solution for the student. The student model may not achieve perfect alignment with the proxy teacher; however, they typically converge closely enough to allow for the substitution of $\mathbf{p}^s$ with its minimizer $\mathbf{p}^s = \mathbf{p}^{t_{px}}$ in Eq.9. This pragmatic approximation is justified by the empirical effectiveness of the approach, as observed in our experiments.
> We agree that establishing theoretical bounds would enhance the credibility of the proposed objective and its implications, we aim to address it in future research.
>
>
> 3. ```The results in Table 1 for all methods are an average over three trials, however, the variation is not presented.```
> **R3**: Thanks for the suggestion. In the initial version, we omitted the variation from Table 1 to maintain clarity, as including variation made each cell overly crowded. Now we have included the variation in appendix A.9 of the updated manuscript.

---

> ### Author Response · Authors · 2023-11-22
>
> 4. ```From Figure 4(b) is not clear why the authors used M=5 in their main experiments. Moreover, it is not discussed in general how to select M or the reasoning for using M>1 given the results in Figure 4(b).```
> **R4**: M is the perturbation order to control the granularity of the perturbation. A larger M means we can add perturbation to the higher-order terms in the polynomial, thus making finer-grained adjustments to the original loss function. The space of the perturbed loss function with a larger M encompasses that with a lower M. To be specific, using M>1 naturally facilitates the more refined adjustments to the original loss function compared to M=1.
>
> The intent of Figure 4(b) is to illustrate the efficacy of the proposed selection methods of the perturbation coefficients, not to justify the specific choice of M>1. The decision to use M=5 in our experiments is underpinned by the goal of achieving a balance between sufficient granularity in perturbation and a manageable search space for the perturbation coefficients.
>
> In the set of experiments of Figure 4(b), we maintained a constant search time of 50 across M values from 1 to 5. This consistency did not yield evidence that M=5 outperforms M=1. However, **increasing the search time** to 100 led to a difference, with M=5 reaching an accuracy of **91.12** compared to M=1's **90.87**.
>
> In practice, we opt for M=5 to ensure a reasonable level of perturbation detail without rendering the search space impractically large.
>
> 5. ```Why using L2 in Figure 3(b) which is consistent with (11), but TVD in Figure 4(a)?```
> **R5**: We follow the settings in (Ren et al., 2022) for Figure 3(b), utilizing the L2 distance as it aligns with their methodology. For Figure 4(a), we use TVD because it provides a more preferred measure of the difference between the probability distributions. Our method is agnostic to the choice of distance measurement.
>
> 6. ```What is used as validation set in the main experiments, Dev in Table 2? Is there a relationship between the size of the validation set relative to the distillation set and performance gain?```
>
> **R6**: In our main experiments, we use a validation set comprising 1000 samples, with RTE being an exception where only 800 samples are used. The validation set is not employed for direct model optimization; instead, we use it for the perturbation coefficient search. An increase in the validation set size can potentially enhance the estimate of the quality score as defined in Eq. 11, leading to a better selection of perturbation coefficients and, consequently, a performance gain of the student model.

---

> > ### Comment · Reviewer_5qNN · 2023-11-22
> >
> > Thanks to the authors for taking the time to respond to the questions and concerns raised in the reviewer. However, the reviewer does not find answers to questions 1 and 2 particularly satisfying, thus the score will remain unchanged.

---

> ### Author Response · Authors · 2023-11-23
> **Follow-up Response 1 to Reviewer 5qNN**
>
> We appreciate the further engagement of the reviewer. We would like to provide more illustration about question 1 and 2, in the expect of addressing the remaining concerns.
>
> ```Why using L2 in Figure 3(b) which is consistent with (11), but TVD in Figure 4(a)?```
>
> - **Background**. Figure 3 involves a synthetic dataset generated from a Gaussian distribution, serves as an illustration experiment. We have included the details in Appendix A.5 and here are some key points: The label $y$ is sampled from an uniform distribution, and $x$ is sampled from $( p(x|y=k) \sim \mathcal{N}(\mu_k, \sigma^2I)$. To calculate the ground truth probability $p^*(y|x) $, we follow that $p^*(y|x) \propto p(x|y)p(y) $. Figure 4(a) involves a real-world text-related dataset, aims to valid the findings obtained from the experiments in Figure 3.
>
> - **Choice of L2 Distance:** On the synthetic Gaussian dataset of Figure 3, L2 distance is used because 1) it aligns with our derivation; and 2) amplifies larger discrepancies between the target and ground-truth distributions. The squaring component of the L2 norm means that larger differences are given more weight, thus 'amplifying' their impact on the overall distance measure.
>
> - **Choice of TVD**. For the real-world dataset of Figure 4, TVD is a suitable measure because it treats all discrepancies between the predicted and ground-truth distributions equally [1]. It provides a straightforward assessment of the total difference without amplifying any single discrepancy, making it a standard measure for evaluating the alignment of probability distributions. We opt TVD here to show that under a real-world distribution distance measure metric, the findings obtained from the synthetic dataset can still hold.
>
> - **Robustness to Distance Metric**. It is important to note that substituting TVD with L2 in Figure 4(a) does not alter our primary conclusions. The observed relationship can be approximated by the equation **y = -0.165x + 0.742**, which aligns with the trend curve in Figure 4(a) and confirms the robustness of our findings across different distance measures.
>
> - **Consistency in Implementation**. Based on Eq.11, we use L2 in the quality score calculation. For coherence and reproducibility, we have consistently applied the L2 distance in our algorithmic implementation. We have uploaded our code about the perturbation coefficients search method. Please check the **.ipynb file in the Supplementary Material** for comprehensive review and verification.
>
> [1] (Ji et al., 2023) "Tailoring Language Generation Models under Total Variation Distance".

---

> ### Author Response · Authors · 2023-11-23
> **Follow-up Response 2 to Reviewer 5qNN**
>
> We appreciate your feedback on our first-round response. In response to your inquiry about question 2, we provide more illustration here.
>
> ```What is used as validation set in the main experiments, Dev in Table 2? Is there a relationship between the size of the validation set relative to the distillation set and performance gain?```
>
> - **Impact of Validation Set Size**. It is well-recognized in machine learning that the size of the validation set relative to the distillation set can affect performance. If the validation set is too small, it may not represent the distribution of the distillation set accurately, leading to suboptimal hyperparameter choices. Conversely, if the validation set is adequately sized, it can provide a reliable basis for model selection and hyperparameter tuning, thus improving the final model performance.
>
> - **Usage of Validation Set in PTLoss**. We would like to clarify that the performance gain mainly comes from the appropriate usage of the validation set, instead of directly incorporating the validation set into the training set for model optimization.In our method, we employ a validation set of moderate scale, not for model training, but to compute a quality score as defined by Eq. 11, which in turn guides the selection of perturbation coefficients. This approach aligns with the commen practice of hyperparameter tuning using a validation set in most machine learning cases, yet our method stands out by being principled and computationally efficient. To make a fair comparison, we also perform hyper-parameter search for each baseline methods in our implementation, as described in the second paragraph of page 8.
>
> - **Limited Size of Validation Set**. In our experiments, we choose the validation set to be reasonably smaller than the distillation set. Specifically, the size of validation set is 1k for all the datasets in the main experiments, with RTE being an exception where only 800 samples are used. In comparison, the size of largest distillation set is 314k. This aligns with our setting that when distilling large language models, it usually does not provide access to the ground-truth data.
>
> - **Validation Set Size Study**. We recognize the importance of further examining the relationship between the relative validation set size and model performance. To this end, we have varied the validation set size to assess its impact on performance, comparing our approach against the second-best baseline method. We observed slight performance improvements as the validation set size increases, for both our method and the baseline. Importantly, the performance differential between them remains consistent across varying the validation set size. Due to the time limit, we will leave a more comprehensive ablation study regarding the relative validation set size in the future version.
> | MNLI | # N_v = 500 | # N_v = 1000 | # N_v = 1500 |
> |---------|-----------------|------------------|------------------|
> | PTLoss   | 90.9          | 91.1           | 91.2           |
> | MetaDistill       | 90.5        | 90.8         | 90.8         |
>
>   | RTE | # N_v = 400 | # N_v = 600 | # N_v = 800 |
>   |---------|-----------------|------------------|------------------|
>   | PTLoss   | 82.8          | 83.2           | 83.5           |
>   | MetaDistill       | 80.8        | 81.1         | 81.3         |
>
>
> We hope our follow-up response provides clarity and addresses your remaining concerns. We would be grateful if you could consider the above response in your evaluation and possibly revise your score. Thank you in advance.

---

### Meta-Review · Area_Chair_fvop · 2023-12-12

**Metareview:**

This paper proposes perturbing the conventional KL-divergence loss used for knowledge distillation in order to reduce the possible distributional shift between the teacher predictions and the ground-truth targets. After rebuttal, it received scores of 556, which lies in the borderline case.

On the one hand, reviewers commented that the idea of perturbing the Maclaurin series of the conventional KL divergence is interesting, and the empirical results show that the proposed loss indeed can improve performance across various datasets. On the other hand, two reviewers still showed concerns about this paper regarding problem formulation, and find some arguments somewhat misaligned with each other and the current state of the paper. Since the paper is in general about how to improve knowledge distillation with perturbed loss, the AC also has another question regarding why not the authors conduct experiments on computer vision tasks as well, which could be an additional plus, besides the NLP tasks considered in the paper.

Overall, the reviewers who gave scores of 5 finally did not increase scores, and after careful consideration, the AC thinks that the flaws slightly outweigh the merits, thus would like to recommend rejection of the paper.

**Justification For Why Not Higher Score:**

The problem formulation and claims made in the paper can be made more clear. This paper is a borderline case. Overall, the AC thinks that the flaws outweigh the merits.

**Justification For Why Not Lower Score:**

N/A

---

### Decision · Program_Chairs · 2024-01-16

Reject